# Ameliorative Effect of Ethanolic Extract of *Moringa oleifera* Leaves in Combination with Curcumin against PTZ-Induced Kindled Epilepsy in Rats: In Vivo and In Silico

**DOI:** 10.3390/ph16091223

**Published:** 2023-08-30

**Authors:** Md. Niyaz Alam, Lubhan Singh, Najam Ali Khan, Yahya I. Asiri, Mohd. Zaheen Hassan, Obaid Afzal, Abdulmalik Saleh Alfawaz Altamimi, Md. Sarfaraj Hussain

**Affiliations:** 1Faculty of Pharmacy, IFTM University, Moradabad 244102, Uttar Pradesh, India; 2Department of Pharmacology, Ram-Eesh Institute of Vocational and Technical Education, Greater Noida 201310, Uttar Pradesh, India; 3Kharvel Subharti College of Pharmacy, Subharti University, Meerut 250005, Uttar Pradesh, India; lubhansingh@gmail.com; 4GMS College of Pharmacy, Shakarpur, Rajabpure, Amroha 244221, Uttar Pradesh, India; alikhan_najam@oo.co.in; 5Department of Pharmacology, College of Pharmacy, King Khalid University, Abha 61421, Saudi Arabia; yialmuawad@kku.edu.sa; 6Department of Pharmaceutical Chemistry, College of Pharmacy, King Khalid University, Abha 61421, Saudi Arabia; mzhapharma@gmail.com; 7Department of Pharmaceutical Chemistry, College of Pharmacy, Prince Sattam Bin Abdulaziz University, Al Kharj 11942, Saudi Arabia; o.akram@psau.edu.sa (O.A.); as.altamimi@psau.edu.sa (A.S.A.A.); 8Lord Buddha Koshi College of Pharmacy, Baijnathpur, Saharsa 852201, Bihar, India; sarfarajpharma@gmail.com

**Keywords:** *Moringa oleifera*, curcumin, neuroprotective, Pentylenetetrazole, oxidative stress

## Abstract

The ameliorative effect of ethanolic extract of *M. oleifera* (MOEE) leaves in combination with curcumin against seizures, cognitive impairment, and oxidative stress in the molecular docking of PTZ-induced kindled rats was performed to predict the potential phytochemical effects of MOEE and curcumin against epilepsy. The effect of pretreatment with leaves of *M. oleifera* ethanolic extracts (MOEE) (250 mg/kg and 500 mg/kg, orally), curcumin (200 mg/kg and 300 mg/kg, orally), valproic acid used as a standard (100 mg/kg), and the combined effect of MOEE (250 mg/kg) and curcumin (200 mg/kg) at a low dose on Pentylenetetrazole was used for (PTZ)-induced kindling For the development of kindling, individual Wistar rats (male) were injected with pentyletetrazole (40 mg/kg, i.p.) on every alternate day. Molecular docking was performed by the Auto Dock 4.2 tool to merge the ligand orientations in the binding cavity. From the RCSB website, the crystal structure of human glutathione reductase (PDB ID: 3DK9) was obtained. Curcumin and *M. oleifera* ethanolic extracts (MOEE) showed dose-dependent effects. The combined effects of MOEE and curcumin leaves significantly improved the seizure score and decreased the number of myoclonic jerks compared with a standard dose of valproic acid. PTZ kindling induced significant oxidative stress and cognitive impairment, which was reversed by pretreatment with MOEE and curcumin. Glutathione reductase (GR) is an enzyme that plays a key role in the cellular control of reactive oxygen species (ROS). Therefore, activating GR can uplift antioxidant properties, which leads to the inhibition of ROS-induced cell death in the brain. The combination of the ethanolic extract of *M. oleifera* (MOEE) leaves and curcumin has shown better results than any other combination for antiepileptic effects by virtue of antioxidant effects. As per the docking study, chlorogenic acid and quercetin treated with acombination of curcumin have much more potential.

## 1. Introduction

Epilepsy is a chronic neurological disorder characterized by abnormal electrical activity in the cerebral neurons. Epilepsy affects approximately 50 million people worldwide. Epilepsy affects all age groups, especially young people in their first two decades of life and the elderly [1]. Epileptogenesis, being a neurodegenerative disease, might be developed due to the generation of oxidative species. The brain continuously requires a high oxygen supply and metabolic energy and may be more susceptible than other organs tooxygen species generated during metabolism. Therefore, the consumption of metabolic energy in the central nervous system due to a convulsion state progresses to an oxidative condition in CNS [2]. There ismuch evidence from clinical and experimental studies thathighlight the involvement of oxidative stress in the development of neurodegenerative diseases [3].The importance of oxidative stress as an essential mechanism for understanding the seizures caused by epilepsy has been extensively acknowledged. However, there is a lack of clear evidence that free radicals are actively involved in physiological processes during the oxidative stress induced by epilepsy [4]. Oxidative stress results in functional cellular disruption and causes cell death via the oxidation of biomolecules, such as proteins, lipids, and nucleotides [5]. As a result, treating epilepsy through the use of non-pharmacological and antioxidant methods that target oxidative stress mightbe effective. In addition, many enzymatic and non-enzymatic processes are functioning, which convert these reactive species into safer molecules, e.g., superoxide dismutase converts superoxide to hydrogen peroxide and molecular oxygen; furthermore, catalase converts H_2_O_2_ into O_2_ (molecular oxygen). In this way, in the body, these antioxidants behave as defense systems and provide natural protection from reactive species [3]. Apart from oxidative stress, nitrosative stress also play a very important role in developing cognitive disorders, as whatever superoxide’s are generated can form peroxynitrite radicals, which are believed to be more reactive substances.

Being an effective scavenger of reactive oxygen species and reactive nitrogen species (ROS andRNS) alleviates lipid peroxidation, DNA damage, etc., and can lead to consequences of seizure development and other cognitive disorders [6,7]. Commonly used anticonvulsant drugs like sodium valproate, phenytoin, phenobarbitone, and carbamazepine treat symptomatic effects, not the underlying pathological state of epilepsy. About 80 percent of epileptic patients are adequately controlled with currently available anticonvulsant drugs, while 20% experience restorative failure and want to continue treatment [8]. The fundamental disadvantage of antiepileptic medications is that they cause undesirable side effects and require long-term adherence during the duration of treatment. Both epilepsy and antiepileptic treatments have negative effects on learning and memory. In the past few years, a large number of newer antiepileptic drugs have been approved or are in their last phase of development as an add-on therapy for poorly controlled epilepsy; however, the safety and tolerability of these drugs need to be proven [9,10]. Although there is no experimental model that could faithfully reproduce all human temporal lobe epilepsy (TLE) features, some models have beenselected to ask a specific set of questions. The Pentylenetetrazole (PTZ) kindling model is widely accepted as an experimental animal model for estimating the effectiveness of antiepileptic drugs or studying the pathogenesis of epilepsy [11].

Kindled seizures have been shown to cause neuronal loss in the limbic systems of carbonic anhydrase 1 (CA1), carbonic anhydrase 3 (CA3), and the dentate gyrus of the hippocampus, the amygdala, and the entorhinal cortex [12]. Memory impairment has been attributed to the hippocampus’s neurological damage. Increased activity of glutamatergic transmission has also been found to play a crucial role in the neuronal cell death of PTZ kindling in rats due to free radical generation [13]. In order to control epilepsy and its consequences, exogenous dietary supplementation (antioxidants) could serve as a beneficial strategy. The origins of modern medications couldbe explored in traditional medicine. We chose two herbal drugs on the basis of their antioxidant properties: one wasthe leaf of *M. oleifera* Lam (Moringaceae), and the other was curcumin (isolated compounds). Both of these plants wereused in African and Indian traditional medicine to treat not only seizures but also leprosy, stroke, anemia, and mental disorders [14,15].

*M. oleifera* Lam. leaves are a good source of nutrition and have anti-tumor, anti-inflammatory, anti-ulcer, anti-atherosclerotic, and anticonvulsant properties due to the presence of polyphenolic and flavonoid compounds such as quercetin, chlorogenic acid, kaempferol, beta-carotene, and amino acids, which contribute to their antioxidant properties [16].The quantitative assessment of quercetin and chlorogenic acid in *M. oleifera* leaves of the ethanolic extract (MOEE) was validated by HPTLC methods and was quantified in our previous studies [17].

One such medicine is curcumin, which has been shown to ameliorate or even prevent the progression of diseases [18]. Turmeric has been used for decades in India for its health advantages as well as itsspice and colorant. Curcumin, a principal curcuminoid in turmeric, is obtained from the dried rhizomes of the plant *Curcuma longa* [19]. Curcumin has been reported to possess antioxidant, anti-inflammatory, anti-proliferative, anti-inflammatory, anticancer, antiepileptic, antidepressant, immunomodulatory, neuroprotective, antiapoptotic, and antiproliferative effects [20]. Curcumin is also an effective scavenger of reactive oxygen species and reactive nitrogen species [21]. The aforementioned properties of curcumin suggest the potential of using it to treat PTZ-induced kindling while preventing seizures and memory loss.

Therefore, the combined effect of curcumin and *M. oleifera* ethanolic extract (MOEE) supplementation on epileptic seizures, cognitive impairment, and oxidative stress in PTZ-induced rats was assessed in the current investigation. The molecular docking investigations aim to provide novel viewpoints on the creation of anti-epileptic drugs.

## 2. Results

### 2.1. Effect of MOEE and Curcumin on the Seizure Severity Score in Pentyletetrazole Treated Rats

The higher the seizure score, the lowerthe seizure protective effect was, and vice versa. After the challenge dose was applied in the kindling, all rats were placed in a group that had lived without any problem. When compared with the PTZgroup, the combined low dose (250 mg/kg) of *M. oleifera* ethanolic extract (MOEE) and low dose (200 mg/kg) of the curcumin-treated groups demonstrated a significant difference in the seizure score. When compared with PTZ groups to standard (valproic acid) groups, they showed a significant difference in the seizure score. The combined low dose of the *M. oleifera* ethanolic extract (MOEE) and a low dose of the curcumin-treated group compared with valproic acid did not significantly vary in the seizure score, which suggests that the combination treatment exerted an excellent antiepileptic activity in reducing seizure activity (Figure 1).

### 2.2. Neurobehavioral Observations

#### 2.2.1. Effects of MOEE and Curcumin on Elevated Plus Maze Apparatus

There was not a noticeable variance in the initial transfer latency between the open arm and the closed arm. However, when the retention transfer latency was examined after 24 h of the original transfer latency, a significant difference was found. When comparing the control group to the PTZ-treated groups, the retention transfer latency was significantly increased (*** *p* < 0.0001) (Figure 2). When compared tothe control group, the combined effect of MOEE and curcumin at doses of 250 mg/kg and 200 mg/kg was not reflected ata significant difference (*** *p* < 0.0001) in the retention transfer latency. When compared with the PTZ groups, the per se groups showed a significant difference (*** *p* < 0.0001).

#### 2.2.2. Effects of MOEE and Curcumin on Passive Avoidance Test

The passive avoidance test measures the animals’ memory and skill. The initial transfer latency does not differ significantly among the groups; however, PTZ-treated groups considerably delayed the passive avoidance paradigm’s retention latency when compared with the PTZ-treated group, according to Tukey’s post hoc analysis. When compared with the control group, the retention latency significantly decreased in the PTZ group (*** *p* < 0.0001). As MOEE and curcumin were combined with PTZ as a pretreatment, the retention delay was significantly increased (** *p* < 0.001) compared with the PTZ group. When compared with PTZ-treated groups, the combined effect of MOEE and curcumin induced a very significant change (*** *p* < 0.0001) in the retention transfer latency (Figure 3).

#### 2.2.3. Effect of MOEE and Curcumin in Open Field Apparatus

The stimulation of the CNS by PTZ wasrevealed by the increase in locomotor activity, which couldbe triggered by a decrease in the brain’s GABA neurotransmitter. However, there was little difference in the locomotor activity of PTZ rats given valproic acid, MOEE at 500 mg/kg, or curcumin at 300 mg/kg. Moreover, there wasno significant difference in the combination of the MOEE and curcumin supplement on the locomotor activity of the animals compared to the PTZ groups (Figure 4).

### 2.3. In-Vivo Antioxidant Activity

The evidence suggests that antioxidants mightreduce the lesions induced by oxidative free radicals in experimental models of epilepsy.

#### 2.3.1. Effects of MOEE and Curcumin on Brain MDA Level

The malondialdehyde (MDA) level was significantly increased (*** *p* < 0.0001) in the brain inPTZ-treated groups compared with the control groups. The combined effect of MOEE and curcumin supplements significantly reduced (** *p* < 0.001) the PTZ-induced peroxidation of lipids in the brain. The levels of MDA in MOEE and curcumin per se the groups differed significantly (*** *p* < 0.0001) compared with the levels in PTZ groups. The MOEE-treated groups, with both a low and high dose, showed a less significant (* *p* < 0.01) difference in the MDA level compared with that of the PTZ-treated groups. The combined effect of MOEE and curcumin showed a significant decrease (** *p* < 0.001) when compared with PTZ groups (Figure 5A).

#### 2.3.2. Effects of MOEE and Curcumin on Brain of Glutathione Level

The brain GSH level showed a highly significant difference in PTZ-treated groups compared with control groups (*** *p* < 0.0001). When compared to PTZ groups with per se groups, the standard valproic acid groups and the combination of curcumin- and MOEE-treated groups showed highly significant (*** *p* < 0.0001) elevations of GSH levels in the brain. Low-dose MOEE-treated groups showed a non-significant (ns) difference, while high-dose MOEE and curcumin showed a significant difference (** *p* < 0.001) compared with the PTZ-treated groups. Hence, the combination of MOEE and curcumin was more effective atrestoring the depleted GSH level in the brain, and this was equally efficacious compared to standard valproic acid (Figure 5B).

#### 2.3.3. Effect of MOEE and Curcumin on Brain of SOD Levels

In SOD levels, when comparing the control group with the PTZ group, the brain superoxide dismutase (SOD) showed a significant decrease (** *p* < 0.001). The per se groups and the combination of MOEE and curcumin-treated groups showed significantly better significance differences (** *p* < 0.001) in the level of SOD compared with the PTZ-treated group. The standard valproic acid, high-dose MOEE, and curcumin groups had a less significant difference (* *p* < 0.01) in the levels of brain GSH compared with the PTZ group. However, the low dose of MOEE did not show a significant difference (ns) in the level of superoxide dismutase in the brain compared with the PTZ control groups (Figure 5C).

#### 2.3.4. Effect of MOEE and Curcumin on Brain Catalase Levels

In PTZ-treated groups, there was a severe depletion of catalase levels in the brain; the level of catalase in the PTZ-treated group showed a very significant difference (*** *p* < 0.0001) from the control group. When compared to the PTZ-treated group and per se groups, the standard valproic acid group and the combined (curcumin and MOEE)-treated groups showed highly significant (*** *p* < 0.0001) elevations in the catalase level of the brain. The curcumin-treated group (both low dose and high dose), as well as high dose MOEE, demonstrated significant elevation (** *p* < 0.001) in the catalase level of the animals compared with PTZ-treated groups (Figure 5D).

#### 2.3.5. Effects of MOEE and Curcumin on Brain of NO Levels

The NO levels were observed to increase remarkably and were highly significant (*** *p* < 0.0001) in comparison with PTZ-treated groups as well as the control groups. The low-dose treatments with MOEE or curcumin showed only a significant difference (** *p* < 0.001), whereas the standard valproic acid group, per se groups, high-dose MOEE or curcumin groups, and the combined MOEE- and curcumin-treated groups showed a highly significant (*** *p* < 0.0001) difference in lowering the nitric oxide level compared to PTZ-treated groups (Figure 5E).

#### 2.3.6. Effects of MOEE and Curcumin in Brain of AchE Levels

The level of acetylcholine in the brain is a good indicator of memory, and acetylcholinesterase activity has also been used as a substitute for acetylcholine. The current study’s findings are consistent with the PTZ group’s raised AchE activity, which showed that elevated AchE was one of the causes of memory loss. The PTZ-treated group showed a highly significant difference (*** *p* < 0.0001) in elevated AchE levels compared with the control groups. However, AchE activity in all the remaining groups showed a highly significant difference (*** *p* < 0.0001) in the decreasing AchE level compared with the PTZ groups (Figure 5F).

### 2.4. Binding Mode Analysis of Curcumin, Quercetin, Chlorogenic Acid and Valproic Acid

To evaluate the dependability and repeatability of the docking process for our investigation, the internal ligand of flavin-adenine dinucleotide (FAD) was docked into the PDB ID: 1RT2 in the current study (Figure 6). The compound “FAD” has a 3.35 root mean square deviation (RMSD) between its anticipated conformation and its actual X-ray crystallographic conformation.

#### 2.4.1. Post-Docking Analysis

The docking analysis was performed on each naturally isolated molecule in the active site of HGR (PDB ID: 3DK9), and valproic acid was docked as a standard ligand. The binding energy (Kcal/mole), inhibitory constant (Ki), and interactions (H-bond) for these compounds are given in Table 1. All the individual interactions of the molecules for 3DK9 are shown in Figure 7a–d, respectively. Through docking experiments, it was shown that curcumin is most effective in the active part of human glutathione reductase (PDB ID: 3DK9).

#### 2.4.2. Binding Mode Analysis

Chlorogenic acid formed five hydrogen bonds with three different amino acids (Figure 7a). The catechol moiety formed three hydrogen bonds with Serine51 (Ser51) at a distance range of 2.05 to 2.10 Å; the carboxylic acid group formed a hydrogen bond withGlycines31 (Gly31) at1.844 Å. The linker oxygen atom completely filled the hydrophobic pocket of the human glutathione reductase active site by forming a hydrogen bond with threonin 57 (Thr57) at a distance of 1.85 Å. At a distance of 2.074 Å, the catechol moiety of quercetin had a hydrogen bond established with the gly158. The carbonyl and hydroxyl groups formed two hydrogen bonds with glutamic acid 50 (Glu 50) and Ser 51 at a distance of 1.8 and 1.9 Å and completely blocked the catalytic binding site of the receptor, which couldbe the reason whyquercetin wasmore potent than chlorogenic acid (Figure 7b).

At a distance of 1.9 Å, Ser51 and the salicylic acid moiety of curcumin formed a hydrogen bond. At distances of 2.2 and 2.3 Å, respectively, the linker carbonyl group made two hydrogen interactions with Ser30 and cysteine58 (Cys58).Due the structural nature of curcumin, which consisted of two bulky groups connected by conjugated bonds, madeit super flexible to completely occupy the hydrophobic and catalytic binding site, and this could be the reason that curcumin was shown to be the most potent among others (Figure 7c). Valproic acid formed three hydrogen bonds with three different amino acids (Figure 7d). At a distance of 1.808 Å, the carbonyl (=O) group establisheda hydrogen connection established with Thr57. The hydroxyl (-OH) group formed two hydrogen bonds with Cys 58 and Ser 30 at a distance of 2.18 Å and 2.11 Å, respectively, which occupied the hydrophobic pockets. As valproic acid is a small structure, and is not able to occupy the whole binding pocket, this couldbe the reason that valproic acid is lacking in providinga good docking score. In our study, molecular docking analysis demonstrated the presence of ligands in the active binding pocket of the target enzyme. The ligands exhibited quite similar interactions, as explained by Cetin et al. 2021 [22], which further elaborated that these interactions might be held responsible for the inhibition of the enzyme. This is the major shortcoming of the molecular docking approach to identify whether the molecule attached tothe target is agonistic or antagonistic in its activity. The possible mechanism of antiepileptic effects or neuroprotection against PTZ was conducted by binding the mode analysis of curcumin, quercetin, and chlorogenic acid. ROS generated by oxidative stress wasthe primary cause of epilepsy, which is an abnormal interruption of nerve cell activity in the brain. ROS are a particular class of oxygen-containing unstable molecules that easily interact with other molecules in a cell. Reactive oxygen species in cells have the potential to harm proteins, RNA, and DNA, which results in cell death. A crucial enzyme in the cellular regulation of reactive oxygen species (ROS) is glutathione reductase (GR). Therefore, activating GR can uplift the antioxidant property, which leads to the inhibition of ROS-induced cell death in the brain. Thus, epilepsy can be prevented after molecular modeling studies and it has been observed that the active pocket or site of human glutathione reductase is “Y”-shaped; to obtaina good binding activity, it needs long and “Y”-shaped ligands and it can also be concluded if any of the compounds between chlorogenic acid and quercetin are treated with the combination of curcumin with much more potential.

## 3. Discussion

The current pharmacotherapy of epilepsy has the limitations of a chronic course, involving unavoidable adverse effects asan economic burden, and falls short of the therapeutic goal of a seizure-free status in nearly one-third of patients [23]. The use of plant-based products for the treatment of convulsions has been a longstanding tradition in Asia, Africa, and South America [24]. Many plant extracts have shown the presence of anticonvulsant activity in animal seizure models, which has been attributed to the action of flavonoids, furanocoumarins, phenylpropanoids, and terpenoids on gamma amino butyric acid (GABA) receptors and voltage-gated ion channels [25]. These phytochemicals facilitate the maintenance of normal physiological function onmajor inhibitory neurotransmitters [26]. Modern laboratory methods and anemphasis on evidence-based medicine have renewed interest in research on herbal items in an effort to discover a safe and effective antiepileptic compound. The present understanding of epilepsy indicates that it mightbe useful to develop antiepileptic medications with added antioxidant action and medicinal plants asan excellent source for such an endeavor. Traditional medicinal plants provide a wealth of information for the creation of contemporary pharmaceuticals. Herbal medicine has drawn extensive appreciation from research bases and enterprises lately at both the national and international levels. Hence, there has been extreme focus with respect to potential phytochemicals to protect neuronal activity and the defensive component against epilepsy.

Epilepsy is one of the most common neurological disorders, which is estimated to affect around 50 million people worldwide and is characterized by epileptic seizures associated with complex molecular, biochemical, physiological, and anatomical changes in the brain [27]. The potential causes of epilepsy include brain injury, a brain tumor, stroke, or inflammation in the brain [28]. Epileptic seizures occur due to the abnormal discharge or excessive firing activity of neurons in the brain [29]. Neuronal death in epilepsy could be attributed to oxidative stress-induced free radical generation in the brain due to lipid peroxidation, protein oxidation, and DNA damage [30]. Both epilepsy and antiepileptic medications have a negative impact on an epileptic patient’s ability to learn and remember things. Currently, available antiepileptic drugs target only the symptoms but do not prevent the underlying pathology of epilepsy or its associated comorbidities [31]. Still, more than 30% of patients experience epileptic seizures after therapy with AED [32,33]. It is, therefore, necessary to find alternative natural remedies (phytoconstituents or nutraceuticals) to traditional AED that might offer beneficial clinical effectiveness and tolerance with low side effects.

*M. oleifera* Lam. (Family: Moringaceae) is a common culinary plant known as the drumstick tree. *M. oleifera* Lam is a widely available plant in Southeast Asia that has been evaluated for the presence of antioxidant activity in a few earlier studies [34]. The most fascinating characteristics of this species are its antioxidant [35] and anti-inflammatory characteristics [34]. Traditional systems of medicine claim to suggest that the leaves of *M. oleifera* have the potential for the treatment of epilepsy [36]. The leaves of *M. oleifera* have been reported to contain a number of phytoconstituents, including alkaloids, carotenoids, flavonoids, polyphenols, phenolic acids, tannins, saponins, and vitamins. The leaves are also said to be rich in vitamins A and C, beta-carotene, chlorogenic acid, kaempferol, and quercetin, which contribute to their antioxidant properties [37,38]. Curcumin, a principal curcuminoid that is present in turmeric, can beobtained from the dried rhizomes of *Curcuma longa* [39].Curcumin has been reported to possess antioxidant, anti-inflammatory, antiepileptic, immunomodulatory, and neuroprotective activity [40]. Curcumin is also believed to be an effective scavenger of reactive oxygen and nitrogen species [41]. Therefore, the characteristics of curcumin offer enormous potential as a medicine for treating PTZ-induced kindling that has seizures and cognitive impairment. Due to the presence of phenolic groups, curcumin acts as a strong antioxidant and inhibits the generation of reactive oxygen species such as superoxide anions and nitrite radicals both in vitro and in vivo [42]. The presence of hydroxyl (OH) groups in phenolic compounds couldcontribute directly to their antioxidant activity and be a significant predictor of their radical scavenging ability [43]. The Pentylenetetrazole (PTZ) kindling model is widely accepted as an experimental animal model with which to evaluate the effectiveness of antiepileptic drugs or study the pathogenesis of epilepsy [44,45]. Kindling is the process of repeatedly decreasing the seizure threshold in the brain by electrical or chemical stimulations, which results in repetitive seizures. It causes the gradual development of seizures, which results in generalized tonic-clonic seizures often associated with cognitive impairments [46]. Pentylenetetrazol’s repeated administration with a sub-convulsive dose determines the nature and intensity of convulsant activity. The seizure activity of drugs in animals is evaluated on the basis of the seizure score gained by the administered drug [47]. The higher the seizure score, the lesser the seizure protective effect, and vice versa. Compared to PTZ-treated (4.83 ± 0.17) animals, low-dose curcumin (3.33 ± 0.21) and low-dose MOEE (3.50 ± 0.50) grouped animals had a significant difference in their seizure score value. The high-dose curcumin-treated (2.66 ± 0.33) groups showed a highly significant difference, whereas the high-dose MOEE-treated (3.00 ± 0.26) groups showed a significant difference in their seizure score compared to the PTZ-treated groups. The combined low-dose MOEE and low-dose curcumin-treated group showed a highly significant difference in their seizure score (2.50 ± 0.34) compared to the PTZ-treated group. The standard (valproic acid)-treated groups hada highly significant difference in their seizure score (2.16 ± 0.31) compared to the PTZ groups. There were significant differences in the convulsion score between the valproic acid groups and the combined low-dose MOEE and low-dose curcumin groups, which suggests that the combination treatment exerted excellent antiepileptic activity in reducing seizure activity. Neurobehavioral assessments were performed to assess cognitive functions, such as learning and memory, by elevated plus maze (EPM) and passive avoidance (PA), as described by Sarangi et al. (2017) [48]. The biological processes in the brain that contribute to impairment in cognitive function have been reported to be influenced by ongoing seizure activity and antiepileptic drug treatment [49]. The reduction in cognitive functions in kindled rats might be caused by an assortment of circumstances. One of several explanations refers to degenerative processes in brain structures related to ischemia and hypoxia [50]. It has been shown that kindled seizures are associated with selective degeneration of cortical and limbic structures, including hippocampus areas, and involve loss of neurons, glial and neuronal growth, and astrocyte hypertrophy [51]. In addition, seizure activity and antiepileptic activity have increased the level of free radicals and reduced antioxidant scavenging defense activity [52]. This imbalance in the body’s oxidant and antioxidant defense mechanisms may result in seizures and cognitive impairment. Elevated plus maze apparatus wasused to evaluate memory in rodents, and there was no significant difference in the initial transfer latency between open and closed arms. On the other hand, a significant difference was seen in the retention transfer latency, which was assessed 24 h following the initial transfer latency. PTZ kindling induced a highly significant increase (*** *p* < 0.0001) in the retention transfer latency compared with the control group, according to post hoc analysis. The dose-dependent effect of curcumin and MOEE inverted the effect of PTZ-induced kindling on retention transfer latency. The retention transfer latency decreased less significantly (* *p* < 0.01) in low-dose curcumin and low-dose MOEE groups compared with the PTZ group. However, when significant amounts of curcumin and MOEE were used, the transfer latency was significantly reduced (** *p* < 0.001) compared with PTZ groups. When compared tothe control group, the combined impact of MOEE and curcumin at doses of 250 mg/kg and 200 mg/kg did not show a highly significant difference (*** *p* < 0.0001) in retention transfer latency on EMP tests. When compared with the PTZ group, per se groups showed a significant difference (*** *p* < 0.0001).The previous study also reported that 200 mg/kg and 300 mg/kg doses did not cause any significant changes in the transfer latency [45]. When Pentylenetetrazole groups were compared with the control group, their retention latency decreased significantly (*** *p* < 0.0001). However, when MOEE and curcumin were combined with PTZ results, there was a substantial dose-dependent increase (** *p* < 0.001) in retention latency compared with PTZ groups. When compared with PTZ-treated groups, the combined effects of MOEE and curcumin resulted in a highly significant difference (*** *p* < 0.0001) in the retention transfer latency of the passive avoidance test. This increase in locomotor activity revealed the stimulant effect of PTZ on the CNS due to decreased GABA neurotransmitters in the brain [53]. Moreover, the kindling process is known to increase the strength of excitatory synaptic connections and decrease the strength of connectivity between inhibitory synapses, which could be the reason for an increase in the locomotor activity of the PTZ groups [54]. However, PTZ groups with 300 mg/kg curcumin, 500 mg/kg MOEE, or the standard (valproic acid) showed no variation in locomotor activity. In addition, there wasno significant change in locomotor activity in the rats treated with MOEE and curcumincompared with the PTZ groups. Free radicals are frequently created in the body as a result of aerobic metabolism. Over production of these free radicals (ROS and RNS) results in damage to lipids, proteins, and DNA in the cells, eventually leading to various neurological disorders [55]. Moreover, the brain is quite susceptible to oxidative damage as it contains a high amount of polyunsaturated fatty acids, which can be readily peroxidized [56]. To counteract excess free radical generation, there are several endogenous antioxidants (catalase, superoxide dismutase, and glutathione) that produce protective effects against free radical generation in the brain.

Lipid peroxidation is a process in which free radicals form and react with lipids present in cell membranes, leading to cell damage [57]. Reactive aldehydes, such as malondialdehyde (MDA), are the end products of lipid peroxidation. MDA is an end product of free radical generation and is used as an indicator of oxidative stress in biological systems [58]. Malondialdehyde levels were found to be significantly higher (*** *p* < 0.0001) in the brains of PTZ-treated groups compared with the control groups. MOEE and curcumin supplementation substantially reduced PTZ-induced lipid peroxidation in the brain (** *p* < 0.001). The per se groups of MOEE and curcumin showed a significant difference (*** *p* < 0.0001) in MDA levels when compared with the PTZ-treated groups. The MOEE groups, with both a low dose and high dose, showed a less significant (* *p* < 0.01) difference in the MDA level compared with PTZ-treated groups. The combined effect of MOEE and curcumin showed that the level of MDA significantly decreased (** *p*< 0.001) when compared with the PTZ-treated groups. The combination supplement showed a similar effect to standard valproic acid (** *p* < 0.001) in reducing the MDA level and, hence, lipid peroxidation in the brain. This antioxidant effect of curcumin in PTZ-kindled rats wassupported by the findings of a recent study [59], where an oral supplement of curcumin decreased catalase, MDA, and glutathione in the rat cerebellum and cerebrum. By scavenging free radicals, GSH served a crucial function in protecting cells from oxidative damage. This was used as an indicator of oxidative stress. The GSH level was significantly lower in the PTZ groups compared with the control group (*** *p* < 0.0001).When compared with the PTZ-treated group, the per se groups, the standard valproic acid group and the combination of the curcumin and MOEE groups and all showed a highly significant (*** *p* < 0.0001) increase in brain GSH levels. Low-dose MOEE-treated groups showed a non-significant (ns) difference, while high-dose MOEE and curcumin showed a significant difference (** *p* < 0.001) compared with the PTZ-treated groups. Hence, the combination treatment group was more effective at restoring depleted GSH levels in the brain and was equally efficacious as standard valproic acid treatment.

The PTZ group had significantly lower brain superoxide dismutase levels compared with the control group (** *p* < 0.001). The per se groups and the combination of MOEE and curcumin-treated groups were far better than the standard valproic acid groups and showed a significant difference (** *p* < 0.001) in the level of SOD compared with the PTZ groups. The standard valproic acid, high-dose MOEE, and curcumin groups had less significant differences (* *p* < 0.01) in their GSH levels compared with the PTZ groups. The brain catalase levels of the PTZ-treated group showed a highly significant difference (*** *p* < 0.0001) whencompared with the control group, as PTZ caused the severe depletion of catalase levels in the brain. When compared to PTZ-treated groups, the per se groups, the standard valproic acid group, and the combined (curcumin and MOEE) treated groups showed a highly significant (*** *p* < 0.0001) elevation in the catalase level of the brain. The nitric oxide levels were observed to be increased remarkably and significantly (*** *p* < 0.0001) in the brain with PTZ-treated groups in comparison with the control. Low-dose treatments with MOEE or curcumin showed only a significant difference (** *p* < 0.001), whereas the standard valproic acid group, the per se groups, high-dose MOEE or curcumin groups, and the combined MOEE and curcumin-treated groups showed a highly significant (*** *p* < 0.0001) difference in lowering nitric oxide levels compared to the PTZ-treated group. The present research results are consistent with enhanced AchE activity in the PTZ group, suggesting that increased AchE was an additional factor in memory loss. When compared to the normal control group, the PTZ-treated group displayed a highly significant difference (*** *p* < 0.0001) in higher AchE levels. However, when compared with PTZ-treated groups, AchE activity in all the other groups showed a highly significant difference (*** *p* < 0.0001).

The above-mentioned results indicate that the combined effect of *M. oleifera* leaves, ethanolic extract, and curcumin experiencesignificant protection against PTZ-induced kindled epilepsy in rats. This could be due to the presence of flavonoids and phenolic compounds. A reduction infree radicals as an increased expression of antioxidant enzymes resulted in a reduction in lipid peroxidation. The majority of plant extracts reduced a significant number of free radicals. Phenolic compounds are unique secondary metabolites that are present in plants and exhibit a number of therapeutic applications, such as antioxidants, anticancer, neuroprotective, etc. The presence of hydroxyl groups contributed significantly to the phenolic compound’s scavenging capacity. According to the results, ethanolic extracts of *M. oleifera* leaves and curcumin hadstrong antioxidant, free radical scavenging, and antiepileptic actions. The possible mechanism of antiepileptic effects or neuroprotection against PTZ through thebinding mode analysis of curcumin, quercetin, and chlorogenic acid epilepsy wascharacterized as an aberrant interruption of nerve cell activity in the brain produced by oxidative stress-induced reactive oxygen species (ROS). ROS are unstable molecules containing oxygen that easily interact with other molecules in a cell. Reactive oxygen species in cells can damage genetic materials and proteins, which can lead to cell death. Glutathione reductase (GR) is an enzyme that regulates reactive oxygen species (ROS) in the cell. 

Therefore, activating GR can uplift the antioxidant property, which leads to the inhibition of ROS-induced cell death in the brain; thus, epilepsy can be prevented. After the molecular modeling studies, it was observed that the active pocket or site of human glutathione reductase was“Y” shaped, and to obtain agood binding activity, it needed long and “Y”-shaped ligands. It couldalso be concluded that if any of the compounds between chlorogenic acid and quercetin weretreated with the combination of curcumin, there would bemuch more potential.

## 4. Materials and Methods

### 4.1. Reagents and Chemicals

All the reagents and chemicals were purchased from authentic resources (Sigma Aldrich, St. Louis, MO, USA). Curcumin was purchased from Sigma Chemical Co. (St. Louis, MO, USA). High-performance chromatography (HPLC) analysis of curcumin powder revealed that it contained 95.02% curcuminoids. Pentylenetetrazole, reduced glutathione, DMSO (dimethyl sulfoxide), DTNB, thiobarbituric acid, tetra ethoxy propane, trichloroacetic acid, pyridine, n-batanol, and sodium dodecyl sulfate were purchased from Sigma Aldrich.

### 4.2. Preparation of M. oleifera Leaves Extract

In the month of January 2020, Moringa oleifera Lam. fresh leaves (Moringaceae) were collected from Moradabad, UP, India, and the specimens were verified (voucher no.: NICAIR/RHMD/Consult/2020/3600-01) and examined by Dr. Sunita Garg, Emeritus Scientist, RHMD, NISCAIR, PUSA Institute, New Delhi 110012, India. The dehydrated leaves of *M. oleifera* material were then ground with a multifunction grinder (Huangdai, China) and fed through a 60-mesh sieve. When using a soxhlet extractor, 100 g of coarsely crushed Moringa oleifera lam leaves were treated with continuous hot extraction at elevated temperatures with ethanol. The ethanolic extract of Moringa oleifera leaves was separately filtered by Whatman filter paper and dried by a rotatory evaporator to obtain thedry extracts. The dry extracts were kept in the freezer (0–4 °C) until they were required. According to dried leaf weight, the extraction yield was 1.36% (*w/w*).

### 4.3. Inducing of Kindled Seizures and Experiment Design

Albino Wistar male adult (150–200 g) rats were used and kept at 25 °C ± 5 °C with a humidity of 50 ± 10% and a 12 h dark cycle and 12 h light cycle. The study received approval from the institutional animal ethical committee (Reg. No. 837/PO/ReBiBt/S/O4/CPCSEA) and was carried out in compliance with the Indian government’s standards on animal guidelines and the Committee for the Purpose of Control and Supervision of Experiments on Animals (CPCSEA).After acclimatization for one week, the animals (Wistar albino male rats) were categorized into ten groups, and each group contained six animals. (Table 2; Figure 1). The selection of the dose of ethanolic extracts of Moringa oleifera leaves (MOEE) was performed based on previously reported studies as well as a survey of the literature [60]. PTZ was dissolved in 0.9% saline and administered intraperitoneally (i.p.) on alternate days ata dosage of 40 mg/kg for a duration of 29 days, whereas curcumin and MOEE were freshly prepared throughout the duration of the study [61]. Curcumin (200 mg/kg and 300 mg/kg, p.o.) [45] MOEE (250 mg/kg and 500 mg/kg, p.o.) [60] and sodium valproate (100 mg/kg, i.p.) wasgiven daily orally (p.o.) for a period of 29 days, 30 min before the PTZ treatment on alternate days, and suspended in 1% carboxymethyl cellulose [45,61]. Animals were housed in a plexiglass chamber (30X24X22 centimeters) after each PTZ dose was administered, and seizure activity was monitored and recorded after 30 min. According to the Racine scale [41], the seizure response’s strength was measured as follows. There werefive possible responses: 0 for no response, l for mouth and facial jerks, 2 for nodding or myoclonic body jerks, 3 for forelimb clonus, 4 for rearing, falling down, forelimb clonus, and 5 for status epilepticus. Figure 8 and Table 2 show the precise experimental layout.

#### 4.3.1. Neurobehavioral Assessment

These behavioral studies were performed 24 h after the completion of the PTZ challenge dose. The elevated plus maze was used to test cognitive impairment after the completion of the PTZ challenge dose. Only one animal was tested at a time for the behavioral study.

#### 4.3.2. Elevated Plus Maze Test

The EPM apparatus consisted of two open arms, two closed arms, and a center region. The elevated plus maze was used to evaluate cognitive impairment in rats, as previously described [62,63]. The initial transfer latency in the first trial was determined by the time needed for the animal to enter with a closed arm while looking away from the central platform with all four limbs. The cutoff time was set at sixty seconds. The rat was then given another 10 s to roam freely through the maze with both open and closed arms. The retention transfer latency test was performed in the same manner as the acquisition trial 24 h later. The elevated plus maze was once again used to confine the rats. On the second trial, the transfer latency was set to sixty seconds.

#### 4.3.3. Step-Through Passive Avoidance Test

Using the previously reported step-through passive avoidance apparatus, memory retention impairment was examined [45,64]. The device had two distinct chambers, each with a steel grid floor. A guillotine door that joined the compartments was present. The black room was kept completely dark, while the white chamber was illuminated by a bulb. For the acquisition trial, each animal was kept in an illuminated chamber. After 60 s, the door of the guillotine between the light and dark chambers was opened for animal habituation, and the initial time of latency was noted when the animal entered the dark chamber. When the initial time was more than 60 s, the rats were not allowed to participate in subsequent experiments. The guillotine door was closed the moment the rat entered the dark room after a 3-s electric foot shock (75 V, 0.2 mA, 50 Hz) was administered through the grid area. Just five seconds later, the rat was taken out of the darkened space and put back into the cage. In the acquisition trial, the retention latency was evaluated after 24 h in the same manner. The delay time was recorded for up to 300 s.

#### 4.3.4. Open Field Test (OFT)

The open field test is an experiment used in scientific research to measure rodents’ levels of general motor activity and fear [65]. It is a widely used quantitative and qualitative assessment of rodents’ general locomotor activity and exploratory desires. The frequency with which behavior in open spaces aligns with common locomotor activity in other contexts 

A rat was placed in the middle of each compartment. Once the test session had begun, care was taken to remain as far away and still as possible if the experimenter plannedto stay in the testing room. Exploratory activity can be dramatically impacted by sudden motion or loudness. Throughout the testing procedure, rodents werefree to roam the chamber. Each line crossed, and thebreak of a photocell beam resulted in one activity point being awarded. The average test time for evaluating an uneasy environmental exploration was five minutes. A 30-min test session wasrecommended if the researcher wanted to look at habituation inan environment that hadbecome more and more familiar. After the test wasover, the rodent was put back in its cage. Rearing behaviors, feces, and grooming activity couldall be assessed in addition to horizontal units of activity.

### 4.4. Measurement of Oxidative Stress

The animals were decapitated, complying with the conclusion of the treatment paradigm and the neurobehavioral assessment: Animals were decapitated under anesthesia (ketamine), their brains were rinsed, and after decapitation, the brains were immediately removed, cleaned with ice-cold saline, and stored at 80 °C until biochemical analysis was conductedwithin seven days. For biochemical estimation, 10% (*w*/*v*) of tissue homogenates weremixed with a 0.1 m phosphate buffer (pH 7.4). The homogenates were centrifuged for 15 min at 10,000× *g* at 4 °C. Supernatants were divided into aliquots, which were then used to make biochemical estimates.

#### 4.4.1. Lipid Peroxidation (MDA)

Malondialdehyde (MDA) was estimated as described previously [66]. In total, 0.1 mL of the sample (homogenate tissue) was mixed with 1.5 mL of thiobarbituric acid (0.8% *w*/*v*), 1.5 mL of acetic acid (20% *v*/*v*), and 0.2 mL of sodium dodecyl sulfate (8.1% *w*/*v*) and heated at 95 °C for 60 min. After the addition of 5 mL of n-butanol/pyridine (15:1) and 1 mL of distilled water, the mixture was cooled using tap water. Following a vortex, the mixture went through a centrifuge at 4000 rpm for a period of ten minutes. The organic layer was separated, and absorbance was measured at 532 nm using a spectrophotometer (Specord 200, Analytic Jena AG, Konrad-Zuse-Straße 1, Jena, Germany).

#### 4.4.2. Reduced Glutathione Estimation

Reduced glutathione (GSH) was measured according to the method of Ellman (1959) [67]. An equal amount of homogenated mixture was mixed with 10% trichloroacetic acid and centrifuged to separate proteins; 0.5 mL of 5′5-dithiobis (2-nitrobenzoic acid); 2 mL of 0.3 M phosphate buffer (pH 8.4); and then 0.4 mL of distilled water were added to the 0.1 mL of supernatant. Within 15 min of vortexing the solution, the absorbance was obtained at 412 nm. 07745 Jena.

#### 4.4.3. Superoxide Dismutase Estimation

Superoxide dismutase activity was accessed according to the method described by Kono, where in a reduction in nitrobluetetrazolium was inhibited by the superoxide dismutase and measured at 560 nm using a Perkin Elmer Lambda 20 spectrophotometer (Norwalk, CT, USA). In a nutshell, the hydroxylamine hydrochloride was added to the combination of the sample and nitrobluetetrazolium to initiate the reaction, and the results were expressed as a unit or milligram of the protein, where one unit of enzyme wasdefined as the amount of the enzyme inhibiting the rate of reaction by 100% [68].

#### 4.4.4. Catalase Estimation

Catalase activity was assayed by the method of Luck, where the breakdown of hydrogen peroxides (H_2_O_2_) wasmeasured at 240 nm [69]. The test solution contained 0.05 mL of the homogenate (10%) supernatant tissue, 3 mL of hydrogen peroxide, and the phosphate-buffered solution, and absorbance was taken at 240 nm. The micromole of H_2_O_2_ decomposed per mg of protein per minute was used.

#### 4.4.5. Nitrite Estimation 

Green and his coworkers’ Greiss reagent, which contained 2.5% phosphoric acid, 0.1% N-(1-naphthyl) ethylenediamine dihydrochloride, and 1% sulfanilamide, wasused in a colorimetric test to determine the development of nitrite in the supernatant, as a sign of nitrite generation. The Greiss reagent and supernatant were mixed in equal parts, and the entire mixture was incubated for 10 min at room temperature. The absorbance was taken at 540 nm. The concentration of nitrite in the supernatant was determined from a sodium nitrite standard curve and expressed as micromoles per liter [70].

#### 4.4.6. Acetyl Cholinesterase Activity

According to Gorun et al., the basic idea underlying thisstrategy wasto determine the amount of thiocholine that wasformed during the hydrolysis of acetylthiocholine. The color was read immediately at 412 nm [71]. A sufficient quantity of ingredients was added to a cuvette containing Ellman’s reagent and a 0.1 M sodium phosphate buffer (pH 8.0). In total, 14.9 mM of acetylthiocholine iodide was added to start the reaction, and the rate of change in absorbance was monitored after two minutes at 412 nm. The results were represented as the nmol substrate hydrolyzed/min/mg protein using 5-mercapto-2-nitrobenzoate’s (13.6X103 M”1 cm’1) molar absorption value.

### 4.5. In Silco Docking Analysis of Curcumin, Quercitin and Chlorogenic Acid in Compare with Valproic Acid as Anti-Epileptic Drug

During the last couple of decades, research on anti-epileptic studies has taken on an impetus in the field of medicinal chemistry. Due to the lack of crystal structures of receptors and selected molecules (curcumin, quercetin, and chlorogenic acid) in a complex form, it was challenging to build the 3D coordinates for further computational analysis. To this end, and to generate more precise and reasonable active site coordinates, molecular docking is conducted by the AutoDock4.2 tool to merge the ligand orientations in the binding cavity. The crystal structure of Human Glutathione Reductase (PDB ID: 3DK9) [72] was retrieved from the RCSB website (https://www.rcsb.org/) on 5 September 2022 [73] and was used to generate initial 3D coordinates because of its high resolution with 1 Å and a ligand in the active site. The macromolecular protein was prepared via a few steps: first, co-crystallized water molecules were deleted along with the addition of polar hydrogen and the computation of the gasteiger charge [74]. The first confirmation of a newly designed molecule, Human Glutathione Reductase, as an active site was produced by superimposing the structure of chosen molecules against a pre-docked ligand in the PDB. The grid box was then determined by the native ligand (FAD) position on the binding site (Gly31, Gly157, Gly158, Glu50, Ala155, and Asp331) [72,75], with XYZ grid points of 60 × 60 × 60 and grid spacing of 0.375 Å. Finally, for docking, grid parameter files (gpf) and docking parameter files (dpf) were written using MGL Tools 1.5.6, and both were carried out with the following parameters: the number of runs: 50, population size: 150, number of evaluations: 2,500,000, and number of generations: 27,000, using the Lamarckian algorithm [76]. All the ligands were optimized by the Avogadro [77] suite program before the docking studies.

### 4.6. Statistical Analysis

Using Graph Pad Prism 8 (San Diego CA, USA), data were examined. The mean and standard error mean (SEM) were used to express the experimental results. A one-way analysis of variance (ANOVA) and Tukey’s multiple comparison tests was used.

## 5. Conclusions

These results indicate that ethanolic extracts of *Moringa oleifera* leaves and curcumin have a substantial anticonvulsant effect against PTZ-induced convulsions. MOEE and curcumin were additionally shown to be protective against PTZ-induced kindling. These effects might be attributed to antioxidant activity, as evidenced by a decrease in MDA, nitrite, and AchE levels and an increase in GSH, SOD, and catalase levels in the brains of PTZ-treated rats. This study indicates that the combined impact of MOEE and curcumin might be a promising natural molecule for use in epilepsy; however, more research is needed to identify the extract’s active ingredients and determine its pharmacodynamic profile. According to docking studies, it has been observed that the active pocket or site of human glutathione reductase is “Y” -shaped, and to obtaina good binding activity, long and “Y” -shaped ligands are needed. It can also be concluded that if any of the compounds between chlorogenic acid and quercetin aretreated with the combination of curcumin, it can have much greaterpotential.

## Figures and Tables

**Figure 1 pharmaceuticals-16-01223-f001:**
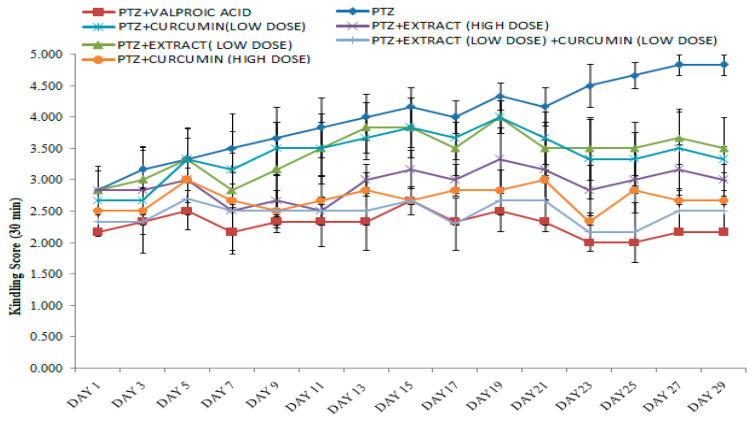
Effect of *M. oleifera* ethanolic extract (MOEE) and curcumin on the seizure severity score in PTZ-treated Wistar albino rats. Each value is expressed as the mean ± SEM; number of animal = 6.

**Figure 2 pharmaceuticals-16-01223-f002:**
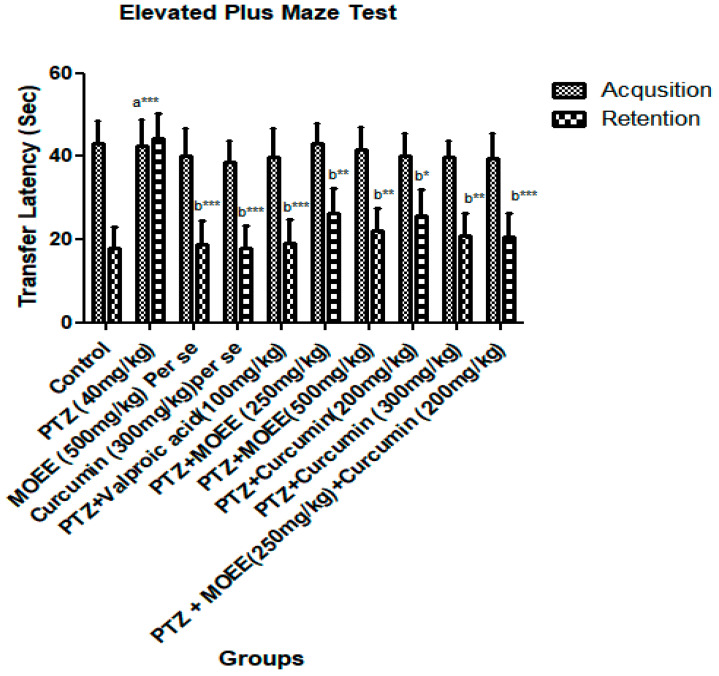
Effect of MOEE and curcumin on elevated plus maze apparatus in PTZ-induced kindled rats. Values areexpressed as the mean ± standard error of the mean (SEM).* *p* < 0.01, ** *p* < 0.001, *** *p* < 0.0001, a—control vs. PTZ, b—PTZ vs. all groups.

**Figure 3 pharmaceuticals-16-01223-f003:**
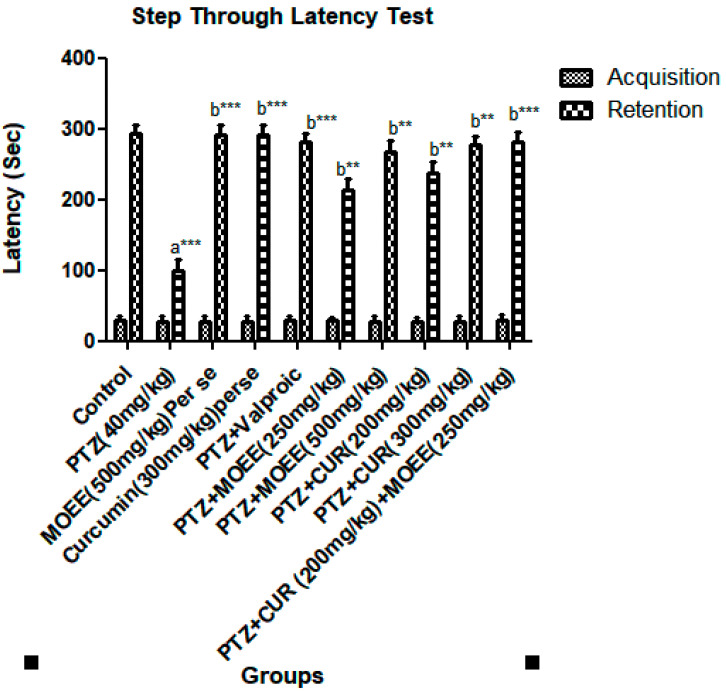
Effect of MOEE and curcumin on the step through latency test in PTZ-induced kindled Rats. Valuesareexpressed as the mean ± standard error of the mean. ** *p* < 0.001, *** *p* < 0.0001, a—control vs. PTZ, b—PTZ vs. all groups.

**Figure 4 pharmaceuticals-16-01223-f004:**
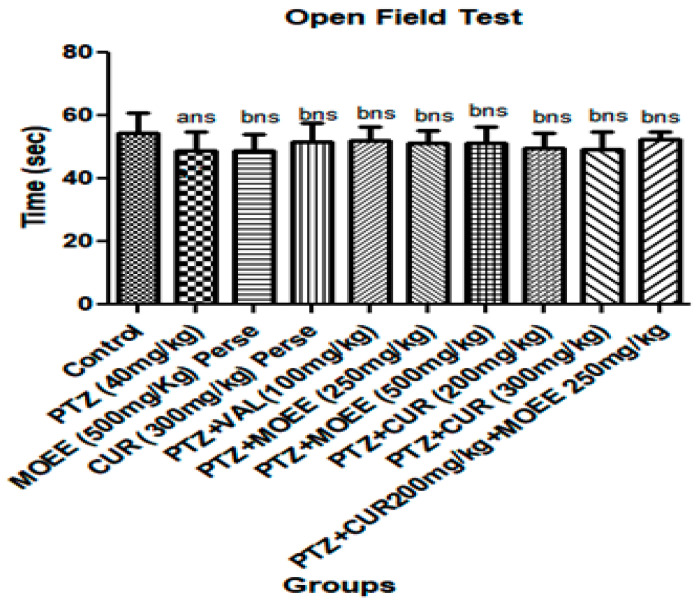
Effect of MOEE and curcumin on open field test in PTZ-induced kindled rats. ns = non-significant, a—control vs. PTZ, b—PTZ vs. all groups.

**Figure 5 pharmaceuticals-16-01223-f005:**
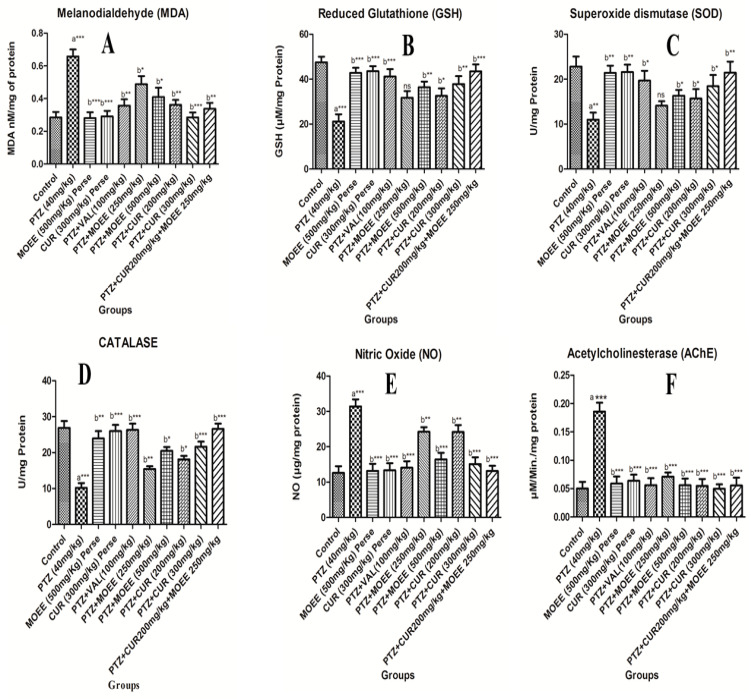
Effect of MOEE and curcumin on thebrain for (**A**) MDA levels, (**B**) Glutathione levels, (**C**) SOD levels, (**D**) Catalase levels, (**E**) Nitric oxide levels, and (**F**) AChE levels in PTZ-induced kindled Rats. Value was expressed as mean ± Standard error of mean (SEM).* *p* < 0.01, ** *p* < 0.001, *** *p* < 0.0001. ns = not significant, a—control vs. PTZ, b—PTZ vs. all groups.

**Figure 6 pharmaceuticals-16-01223-f006:**
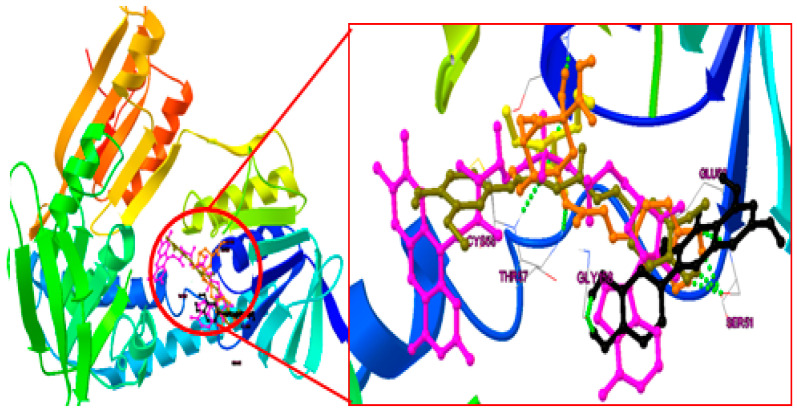
The overlapping of the whole ligand [chlorogenic acid (Saffron color); quercetin (black); curcumin (olive green); internal ligand (pink); valproic acid (Yellow)] in the active site of human glutathione reductase (PDB ID: 3DK9).

**Figure 7 pharmaceuticals-16-01223-f007:**
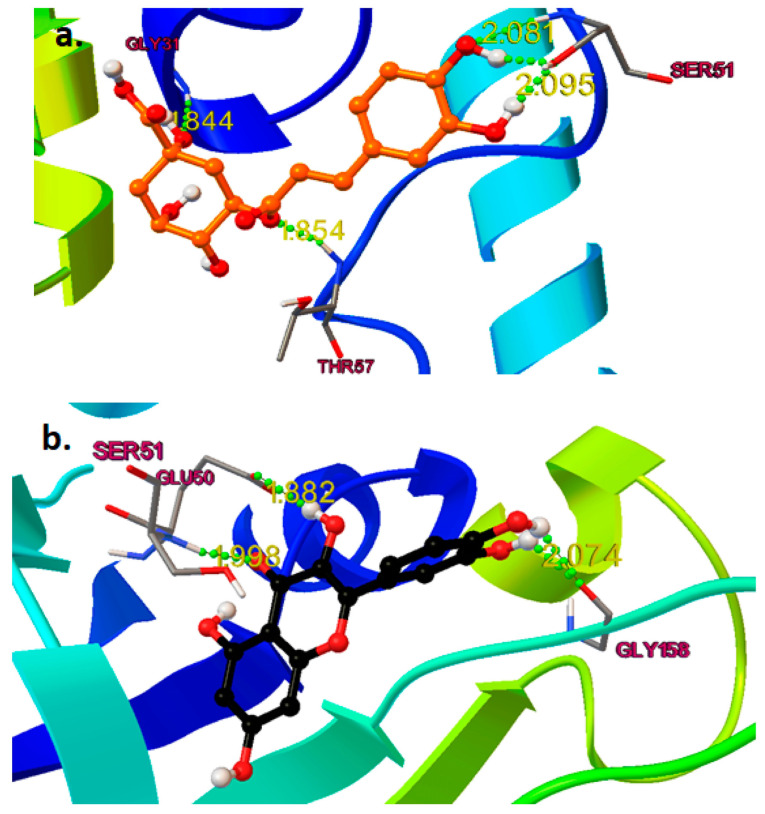
(**a**).Binding interaction of chlorogenic acid (saffron color) in the active site of human glutathione reductase (PDB ID: 3DK9); (**b**)—Binding interaction of quercetin (black); (**c**)—Curcumin (olive green); (**d**)—Valproic Acid (yellow) in the active site of human glutathione reductase.

**Figure 8 pharmaceuticals-16-01223-f008:**
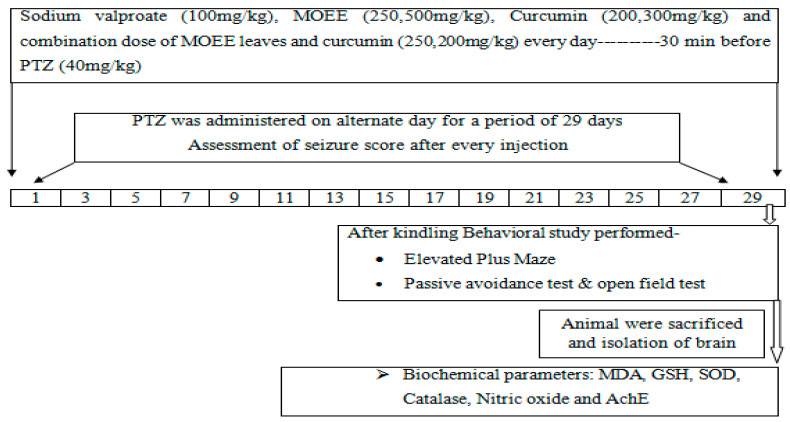
Experimental protocol of Pentylenetetrazole (PTZ)-induced kindling.

**Table 1 pharmaceuticals-16-01223-t001:** List of extracted molecules with their binding affinity in the active site of Human Glutathione Reductase (PDB ID: 3DK9).

Molecule	KI Value	Docking Score (Kcl/mol)	Hydrogen Bond Interaction
Curcumin	0.264 μM	−8.97	SER 51, GLY 29, CYS 58
Quercetin	2.33 μM	−7.68	GLU 50, SER 51, GLY 158
Chlorogenic acid	3.29 μM	−7.48	GLY 31, THR 57, SER 51
Valproic Acid	190.91 μM	−5.07	SER 30, THR 57, CYS 58

**Table 2 pharmaceuticals-16-01223-t002:** Animal grouping and treatments for extracts of *Moringa olifera* leaves and curcumin in PTZ-induced kindled epilepsy.

Groups (n = 6)	Treatments
I	Normal control with vehicle
II	PTZ (40 mg/kg, i.p.) was administered on alternate day for a period of 29 days
III	PTZ (40 mg/kg, i.p.) + MOEE (250 mg/kg, p.o.)—for a period of 29 days and 30 min. before PTZ treatment.
IV	PTZ (40 mg/kg, i.p.) + MOEE (500 mg/kg, p.o.)—for a period of 29 days and 30 min. before PTZ treatment.
V	PTZ (40 mg/kg, i.p.) + Curcumin (200 mg/kg, p.o.)—for a period of 29 days and 30 min. before PTZ treatment.
VI	PTZ (40 mg/kg, i.p.) + Curcumin (300 mg/kg, p.o.)—for a period of 29 days and 30 min. before PTZ treatment
VII	PTZ (40 mg/kg, i.p.) + Curcumin (200 mg/kg, p.o.) + MOEE (250 mg/kg, p.o)—for a period of 29 days and 30 min. before PTZ treatment.
VIII	PTZ (40 mg/kg, i.p.) + Valproic acid (100 mg/kg, i.p.)—for a period of 29 days and 30 min. before PTZ treatment.
XI	MOEE (500 mg/kg, p.o.) per se—for a period of 29 days
X	Curcumin (300 mg/kg, p.o.) per se—given daily by oral route for a period of 29 days

MOEE—*Moringa oleifera* ethanolic extracts.

## Data Availability

The data presented in this study are available on request from the corresponding authors. The data are not publicly available due to the consent provided by participants for the use of confidential data.

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
