# Peer review of "Ameliorative Effect of Ethanolic Extract of Moringa oleifera Leaves in Combination with Curcumin against PTZ-Induced Kindled Epilepsy in Rats: In Vivo and In Silico"

_pharmaceuticals, 2023, doi:10.3390/ph16091223_

Round 1

Reviewer 1 Report

Dear Authors,

The manuscript can be accepted after addressing the below mentioned corrections.

1. The introduction should describe in more detail the relationships between the development of epilepsy and oxidative stress and the antioxidant defense system and the nitric oxide system. The effect of curcumin on these systems should also be characterized.

2. I recommend that the authors include additional references to the literature on the mentioned topic, in particular:  a) Dhir, A. (2018). https://doi.org/10.1002/ptr.6125; b) Forouzanfar, F., et al. (2021).https://doi.org/10.1007/978-3-030-56153-6_21; c) Kaur, H., et al. (2015). https://doi.org/10.1016/j.neuint.2015.07.009; d) Fujisawa, S., et al. (2004). Cytotoxicity, ROS-generation Activity and Radical-scavenging Activity of Curcumin and Related Compounds;  e) Melekh, B., et al. (2017). DOI: 10.7324/JAPS.2017.71013; f) Mishchenko, M., et al.(2022). https://doi.org/10.3390/scipharm90030056;  

3. The quality of figures 1-5 must be improved.

4. Grammatical errors were found in the manuscript. It should be corrected. 

The manuscript needs moderate English editing.

Author Response

RESPONSE SHEET

Manuscript ID: pharmaceuticals-2466534 Type of manuscript: Article

Title: Ameliorative Effect of Ethanolic Extract of Moringa oleifera Leaves in Combination with Curcumin against PTZ Induced Kindled Epilepsy in rats; In Vivo and In Silico Study

Authors: Md. Niyaz Alam*, Lubhan Singh, Najam Ali Khan, Yahya I. Asiri, Mohd. Zaheen Hassan, Obaid Afzal, Abdulmalik Saleh Alfawaz Altamimi, Md. Sarfaraj Hussain.

We would like to extend our sincere thanks to the editor/reviewers for their comments and suggestions. The manuscript has been revised as per their advice and has been highlighted with red color in the manuscript. One more author (Anurag Agrawal) has been added into the manuscript as he contributed significantly in the software validation during the research activities. We believe that addressing all the comments of the reviewers have substantially improved the quality of the manuscript. Detailed point to point replies to the comments are given below.

Comments of Reviewer # 01

Dear Authors,

The manuscript can be accepted after addressing the below mentioned corrections.

Comments 1.

The introduction should describe in more detail the relationships between the development of epilepsy and oxidative stress and the antioxidant defense system and the nitric oxide system. The effect of curcumin on these systems should also be characterized.

Answer: Description has been incorporated in the introduction portion of the manuscript.

Comments 2.

I recommend that the authors include additional references to the literature on the mentioned topic, in particular: a) Dhir, A. (2018).

https://doi.org/10.1002/ptr.6125; b) Forouzanfar, F., et al. (2021).https://doi.org/10.1007/978-3-030-56153-6_21; c) Kaur, H., et al. (2015). https://doi.org/10.1016/j.neuint.2015.07.009; d) Fujisawa, S., et al. (2004). Cytotoxicity, ROS-generation Activity and Radical-scavenging Activity of Curcumin and Related Compounds; e) Melekh, B., et al. (2017). DOI: 10.7324/JAPS.2017.71013; f) Mishchenko, M., et al.(2022).

https://doi.org/10.3390/scipharm90030056;

Answer: These references have been included as per their suitability in the manuscript.

Comments 3.

The quality of figures 1-5 must be improved.

Answer: Correction has been done as suggested.

Comments 4.

Grammatical errors were found in the manuscript. It should be corrected.

Answer: Correction has been done as suggested.

Reviewer 2 Report

This manuscript explored the ameliorative effect of ethanolic extract of M. oleifera (MOEE) leaves in combination with curcumin against seizures, cognitive impairment, and oxidative stress in PTZ-induced kindled rats with molecular docking experiments to predict the potential phytochemical effects of MOEE and curcumin against epilepsy.

A few concerns for the authors.

1.       Figure 3, we have “PTZ+CUR (500 mg/kg)”, it looks like all other figures and the experiment session all indicated it should be “PTZ+CUR (300 mg/kg)”.

2.       Figures 5, 6 ,7 and 8 were not completely uploaded to the text. Therefore, it is hard to evaluate any one of them.

3.       The authors claimed that the possible mechanism of MOEE against oxidative stress was through activating glutathione reductase whose activity can be monitored. It would be more convincing to provide these kinds of data.

4.       The molecular modeling studies provided by the authors only showed that quercetin, a flavonol, and chlorogenic acid, a polyphenolic compound, exhibited good binding affinity towards glutathione reductase by interacting with amino acid residues in the active site of the enzyme. However, all the calculations could not predict whether these kinds of binding would activate the enzyme or inhibit the enzyme. Glutathione reductase has been found to be inhibited by some flavonoids.

5.       Again, with the molecular modeling data provided by the authors, the interaction between curcumin, quercetin, or chlorogenic acid with glutathione reductase should be confirmed by bioassay data.

Author Response

Comments of Reviewer # 02

Comments and Suggestions for Authors

This manuscript explored the ameliorative effect of ethanolic extract of M. oleifera (MOEE) leaves in combination with curcumin against seizures, cognitive impairment, and oxidative stress in PTZ-induced kindled rats with molecular docking experiments to predict the potential phytochemical effects of MOEE and curcumin against epilepsy.

A few concerns for the authors.

Comments 1.

Figure 3, we have “PTZ+CUR (500 mg/kg)”, it looks like all other figures and the experiment session all indicated it should be “PTZ+CUR (300 mg/kg)”.

Answer: Correction has been done as suggested. Comments 2.

Figures 5, 6 ,7 and 8 were not completely uploaded to the text. Therefore, it is hard to evaluate any one of them.

Answer: Again, figures have been uploaded in the manuscript at the required position.

Comments 3.

The authors claimed that the possible mechanism of MOEE against oxidative stress was through activating glutathione reductase whose activity can be monitored. It would be more convincing to provide these kinds of data.

Answer: We claimed the protective mechanism of MOEE against oxidative stress was through activating glutathione reductase on the basis of biological activity performed in our study. However, the mechanism of action of selected natural ligands was just predicted via molecular docking study.

Comments 4.

The molecular modeling studies provided by the authors only showed that quercetin, a flavonol, and chlorogenic acid, a polyphenolic compound, exhibited good binding affinity towards glutathione reductase by interacting with amino acid residues in the active site of the enzyme. However, all the calculations could not predict whether these kinds of binding would activate the enzyme or inhibit the enzyme. Glutathione reductase has been found to be inhibited by some flavonoids.

Answer: This is the shortcoming of molecular docking approach to determine whether attached molecule is agonist or antagonist; however we tried to decipher the mechanism of action of the Quercetin, Chlorogenic acid and Curcumin has been incorporated in the molecular docking study portion of discussion part of manuscript.

Comments 5.

Again, with the molecular modeling data provided by the authors, the interaction between curcumin, quercetin, or chlorogenic acid with glutathione reductase should be confirmed by bioassay data.

Answer: We didn’t perform any in-vitro assay on above mentioned molecules. In current study, we demonstrated the combined effect of curcumin and other natural ligands present in the MOEE. However, Tsuchiya et al., 2014: http://dx.doi.org/10.1271/bbb.60.765 and Choi et al., 2003 : 10.1016/j.ejphar.2003.09.067 elaborated the protective impact of chlorogenic acid and quercetin in rats respectively.

Round 2

Reviewer 2 Report

The revised manuscript addressed most of the concerns.

One question about the Ki values in Table 1 on page 11: Are these Ki values calculated or real values from literatures?